# PRC2 specifies ectoderm lineages and maintains pluripotency in primed but not naïve ESCs

Yongli Shan[1,2,3], Zechuan Liang[1,2,3], Qi Xing[1,3,4], Tian Zhang[1,2,3], Bo Wang[1,2,3], Shulan Tian[5], Wenhao Huang[1,3], Yanqi Zhang[1,2,3], Jiao Yao[1,2,3], Yanling Zhu[1,2,3], Ke Huang[1,3], Yujian Liu[1,3], Xiaoshan Wang [1,2,3], Qianyu Chen[1,3], Jian Zhang[1,3], Bizhi Shang[1,3], Shengbiao Li[1,3], Xi Shi[1,3], Baojian Liao[1,3], Cong Zhang[1,2,3], Keyu Lai[1,3], Xiaofen Zhong[1,3], Xiaodong Shu[1,3], Jinyong Wang[1,3], Hongjie Yao[1,3], Jiekai Chen [1,3], Duanqing Pei[1,3] & Guangjin Pan[1,3]

Polycomb repressive complex 2 and the epigenetic mark that it deposits, H3K27me3, are evolutionarily conserved and play critical roles in development and cancer. However, their roles in cell fate decisions in early embryonic development remain poorly understood. Here we report that knockout of polycomb repressive complex 2 genes in human embryonic stem cells causes pluripotency loss and spontaneous differentiation toward a meso-endoderm fate, owing to de-repression of BMP signalling. Moreover, human embryonic stem cells with deletion of *EZH1* or *EZH2* fail to differentiate into ectoderm lineages. We further show that polycomb repressive complex 2-deficient mouse embryonic stem cells also release Bmp4 but retain their pluripotency. However, when converted into a primed state, they undergo spontaneous differentiation similar to that of hESCs. In contrast, polycomb repressive complex 2 is dispensable for pluripotency when human embryonic stem cells are converted into the naive state. Our studies reveal both lineage- and pluripotent state-specific roles of polycomb repressive complex 2 in cell fate decisions.

[1] CAS Key Laboratory of Regenerative Biology, Joint School of Life Sciences, Guangzhou Institutes of Biomedicine and Health, Chinese Academy of Sciences, Guangzhou Medical University, Guangzhou 511436, China. [2] University of Chinese Academy of Sciences, Beijing 100049, China. [3] Guangdong Provincial Key Laboratory of Stem Cell and Regenerative Medicine, South China Institute for Stem Cell Biology and Regenerative Medicine, Guangzhou Institutes of Biomedicine and Health, Chinese Academy of Sciences, Guangzhou 510530, China. [4] Institute of Health Sciences, Anhui University, Hefei 230601, China. [5] Department of Health Sciences Research, Division of Biomedical Statistics and Informatics, Mayo Clinic, 200 1st Street SW, Rochester, MN 55905, USA. Yongli Shan and Zechuan Liang contributed equally to this work. Correspondence and requests for materials should be addressed to G.P. (email: pan_guangjin@gibh.ac.cn)

Polycomb repressive complexes (PRCs) formed by polycomb group proteins play essential roles in development by mediating chromatin modification[1–5]. The polycomb repressive complex 2 (PRC2 complex) catalyzes histone H3 lysine 27 tri-methylation (H3K27me3) through its core components EZH1, EZH2, EED and SUZ12[6–10]. In contrast, PRC1 contains RING1A and RING1B, E3 ubiquitin ligases that mono-ubiquitinylate histone H2A at lysine 119 (H2AK119ub1)[11, 12]. PRC1 and PRC2 coordinately mediate transcriptional repression through H3K27me3 modification. PRC2 is recruited to specific genomic locations and catalyzes deposition of H3K27me3, which in turn recruits PRC1, thus resulting in generation of H2AK119ub1[13–15]. Whole-genome studies have revealed that PRC2 and its mark H3K27me3 occupy critical developmental genes in both human and mouse embryonic stem cells (ESCs)[2, 3]. Paradoxically, most genes occupied by H3K27me3 are also modified by H3 lysine 4 tri-methylation (H3K4me3)[16–18], thus marking these loci with bivalent modifications to keep lineage genes in a poised state capable of responding rapidly to differentiation cues. Furthermore, these bivalent modifications are rapidly resolved during lineage specification to ensure the proper expression of lineage-specific genes[19–21].

Loss-of-function studies on individual components of PRC2 have been performed and have been reported in *Drosophila* and mice[10, 22–26]. Deletion of three core PRC2 components (*Eed, Suz12* and *Ezh*2) in mice results in severe defect in gastrulation[23, 25, 27, 28], presumably because of the mis-expression of lineage-specific genes[24, 25, 29, 30]. Interestingly, deletion of PRC2 in mouse ESCs exhibits quite different phenotypes. For example, mouse ESCs (mESCs) with *Suz12, Eed* or *Ezh2* deletion appear to be normal with little effect on self-renewal and morphology[6, 7, 31–33]. Transcriptionally, only a small subset of PRC2 target genes are affected in those mESCs. However, *EZH2*[−/−] human ESCs (hESCs) have severe defects in self-renewal and differentiation[34]. Therefore, although PRC1 and PRC2 have been proposed to play critical roles in embryonic development, their exact roles in different pluripotent states (naive vs. primed) and in the differentiation toward three germ layers remain unresolved. In this report, we generated a panel of hESC lines with deletion of *EZH1/EZH2*, *EED* and *SUZ12* and found that these cells underwent spontaneous differentiation to the meso-endoderm germ layers at the expense of the neural ectoderm. Furthermore, we found that PRC2 is required for maintaining pluripotency in only the primed state but not in the naive state.

## Results

**PRC2 is required for pluripotency in hESCs.** To gain insights into the role of PRC2 in cell fate decisions, we generated *EED-, SUZ12-, EZH1-* and *EZH2*-knockout hESC lines by using CRISPR/Cas9 genome editing combined with homologous recombination (Fig. 1a)[35–37]. To ensure that the function of each PRC2 component was completely eliminated, we designed a targeting strategy to ablate the entire functional domain critical for each factor (Fig. 1a, b)[7, 31, 38]. We also included multiple hESC lines in our analysis to rule out the possible variations between cell lines (Fig. 1b). gRNA/Cas9 and the designed targeting vector were electroporated into H1 or H9 hESCs, and the positive clones were subsequently selected by neomycin or puromycin under defined culture conditions (Fig. 1b). Individual surviving colonies after drug selection were then isolated, expanded and analyzed by genomic PCR using different primer combinations shown in Fig. 1a. Using CRISPR/Cas9 combined with homologous recombination, we obtained a significantly high efficiency of homozygous gene targeting, varying from 10–40% across genes or

cell lines (Fig. 1b). Thus, we successfully obtained multiple hESC clones with homozygous deletions of each individual core component of PRC2 (Fig. 1c, d, Supplementary Fig. 1a–e). To prevent potential batch variation between different hESC clones, we used at least two clones with homozygous targeting for each gene for further characterization.

We first observed that hESCs with deletion of different PRC2 components exhibited distinct morphology (Fig. 1c, d, Supplementary Fig. 1c). Whereas *EZH1*[−/−] H1 or *EZH2*[−/−] H1 hESCs remained undifferentiated and morphologically indistinguishable from wild-type (WT) H1 cells (Fig. 1c), *EED*[−/−] or *SUZ12*[−/−] hESCs (H1 or H9 cells) exhibited a gradual and spontaneous differentiation phenotype (Fig. 1c, Supplementary Fig. 1c). Because EZH1 and EZH2 are functionally redundant[7, 39], we performed an additional round of gene targeting in *EZH1*[−/−] H1 or *EZH2*[−/−] H1 cells by removing the antibiotic cassette removed by using Cre-LoxP (see Methods) to generate hESCs with double deletion of *EZH1* and *EZH2* (*EZH1*[−/−]/*EZH2*[−/−] H1). Regardless of whether the double knockout hESCs were originally generated from *EZH1*[−/−] H1 (hereafter denoted *EZH1*[−/−]/ *EZH2*[−/−] H1) or from *EZH2*[−/−] H1 (denoted *EZH2*[−/−]/*EZH1*[−/−] H1) cells, they exhibited gradual differentiation, as observed in *EED*[−/−] H1 or *SUZ12*[−/−] H1 cells (Fig. 1c, Supplementary Fig. 1a, b, g–i). Together, these results suggested that PRC2 is essential for maintaining an undifferentiated state in hESCs.

As expected, H3K27me3 modification on chromatin was completely abolished in *EED*[−/−] H1, *SUZ12*[−/−] H1, and *EZH2*[−/−]/*EZH1*[−/−] H1 cells (Fig. 1e). In contrast, *EZH1*[−/−] H1 or *EZH2*[−/−] H1 cells that maintained an undifferentiated state (Fig. 1c) retained their H3K27me3 modifications, albeit at decreased levels compared with those in WT H1 cells (Fig. 1e). Known pluripotency marker genes such as *OCT4, SOX2* and *NANOG* were inactivated in *EED*[−/−] H1, *SUZ12*[−/−] H1 and *EZH2*[−/−]/ *EZH1*[−/−] H1 cells (Fig. 1f, Supplementary Fig. 1f, j). These results demonstrated that PRC2-mediated H3K27me3 modification is required for maintaining hESCs in a pluripotent state.

**PRC2**[−/−] **hESCs differentiated to default meso-endoderm fate.** During gene targeting, hESCs with homozygous deletion of *EED* or *SUZ12* or double deletion of both *EZH2* and *EZH1* were isolated and further cultured under defined conditions suitable for hPSCs. However, these cells subsequently underwent spontaneous differentiation, as indicated by the loss of typical hESC morphology and alkaline phosphatase (ALP) activity (Fig. 2a, Supplementary Fig. 2a). After examining the markers for the three germ layers using qRT-PCR, we found that these cell lines consistently expressed high levels of meso-endoderm genes but not neural ectoderm genes (Fig. 2b, Supplementary Fig. 2b). As controls, H1 cell-derived embryonic bodies (EBs) expressed genes corresponding to all selected lineages from the three germ layers (Fig. 2b). To further confirm the lineage fate of these differentiated cells, we performed whole-genome transcriptome analysis on *EED*[−/−] H1, *SUZ12*[−/−] H1 or *EZH2*[−/−]/*EZH1*[−/−] H1 hESCs as well as *EZH1*[−/−] H1 and *EZH2*[−/−] H1 hESCs. Spearman's rank correlation analysis on the global transcriptome clearly showed a more closely related differentiation phenotype among *EED*[−/−] H1, *SUZ12*[−/−] H1 and *EZH2*[−/−]/*EZH1*[−/−] H1 hESCs, as compared with that of the WT H1 cells (Fig. 2c, Supplementary Fig. 2c). Moreover, *EZH1*[−/−] H1 or *EZH2*[−/−] H1 cells were similar to the undifferentiated H1 hESCs (Fig. 2c, Supplementary Fig. 2c). Furthermore, *EED*[−/−] H1, *SUZ12*[−/−] H1 or *EZH2*[−/−]/*EZH1*[−/−] H1 hESCs did not express the selected marker genes indicative of pluripotency or neural ectoderm lineage (Fig. 2d)[40, 41]. These data suggested that disruption of PRC2 in hESCs leads to a default differentiation toward the

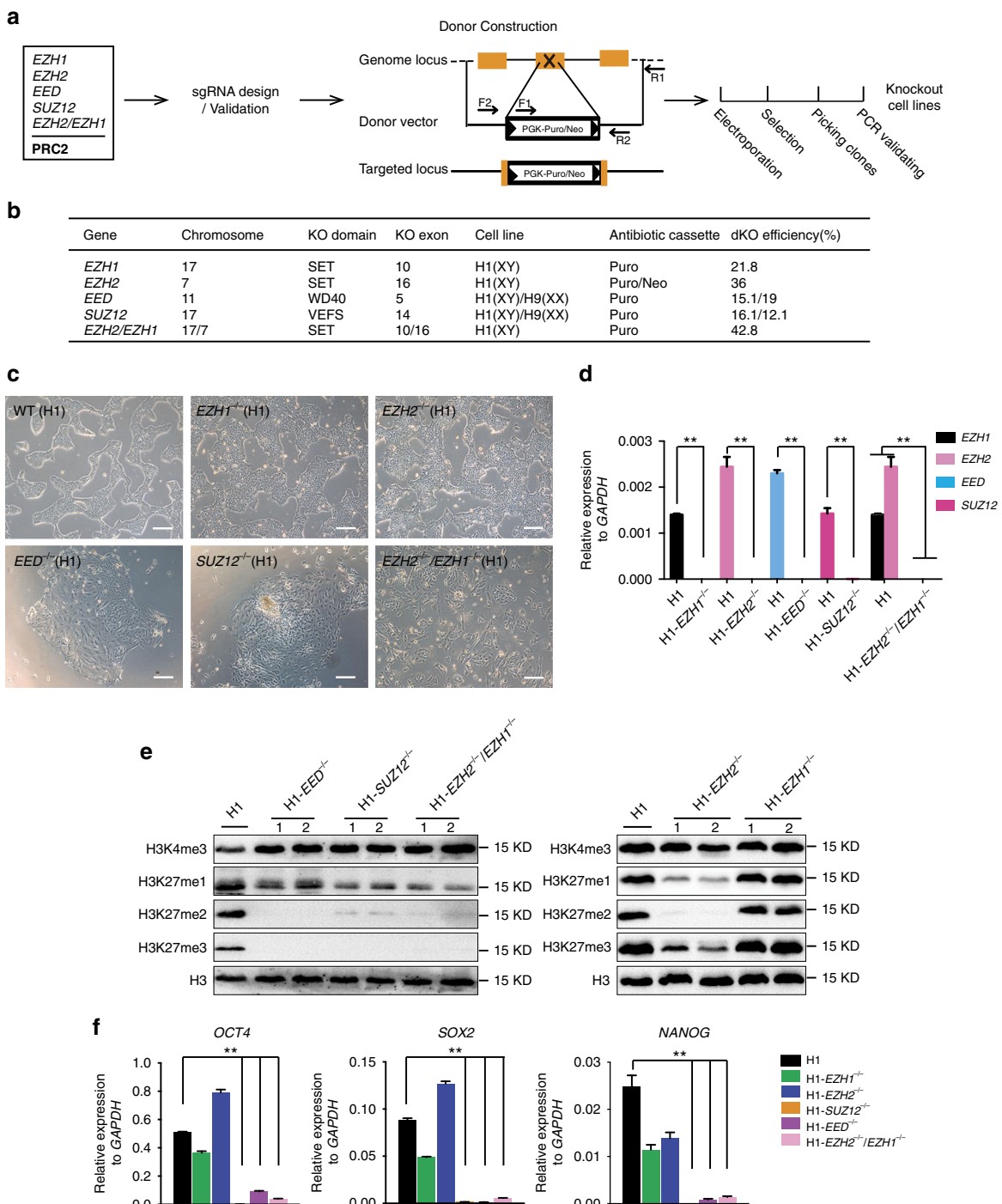

**Fig. 1** Deletion of polycomb repressive complex 2 in human embryonic stem cells. **a** Overview of the gene targeting strategy. gRNA was designed and validated for each polycomb repressive complex 2 (PRC2) component gene showing in *box*. To delete the critical domain for each factor, a homologous targeting vector containing puromycin or neomycin resistant cassette was constructed according to each gene. gRNA/Cas9 together with targeting vector were electroplated into H1 or H9 human embryonic stem cells (hESCs) and selected by the corresponding drug in defined condition. Positive clones were then isolated and expanded for further characterizations. **b** Targeting efficiencies of each gene. The functional domain that was deleted in each factor was shown. For *SUZ12* and *EED*, gene targeting was performed in both H1 and H9 hESCs. **c** Morphology of H1 hESCs with targeted deletion of each gene. *Scale bar*, 200 μm. **d** qRT-PCR analysis on the expression level of each indicated gene in gene targeted hESCs. Wild-type H1 hESCs serve as control. Significance level was determined using unpaired two-tailed Student's *t* tests. **, *P* < 0.01. The data represent mean ± SD from three biological repeats. **e** Total level of the indicated histone modification in gene targeted cells. The total histone modification level was analyzed by western-blot using the specific antibody on the whole-cell lysates from each indicated cell line. **f** qRT-PCR analysis on the expression level of the pluripotent genes, *OCT4, SOX2, NANOG* in gene targeted hESCs. Wild-type H1 hESCs serve as control. Significance level was determined using unpaired two-tailed Student's *t* tests. **, *P* < 0.01. The data represent mean ± SD from three biological repeats. See also Supplementary Fig. 1

meso-endoderm fate. Indeed, two well-known early mesoderm markers, (CALPONIN) and endoderm (SOX17), were detected at high levels by immunostaining in $EED^{-/-}$ H1, $SUZ12^{-/-}$ H1 or $EZH2^{-/-}/EZH1^{-/-}$ H1 hESCs but not in WT, $EZH1^{-/-}$ or

$EZH2^{-/-}$ H1 hESCs (Fig. 2e, Supplementary Fig. 2d–f). Together, these data demonstrated that PRC2 regulates the loss of pluripotency by suppressing the meso-endoderm differentiation program.

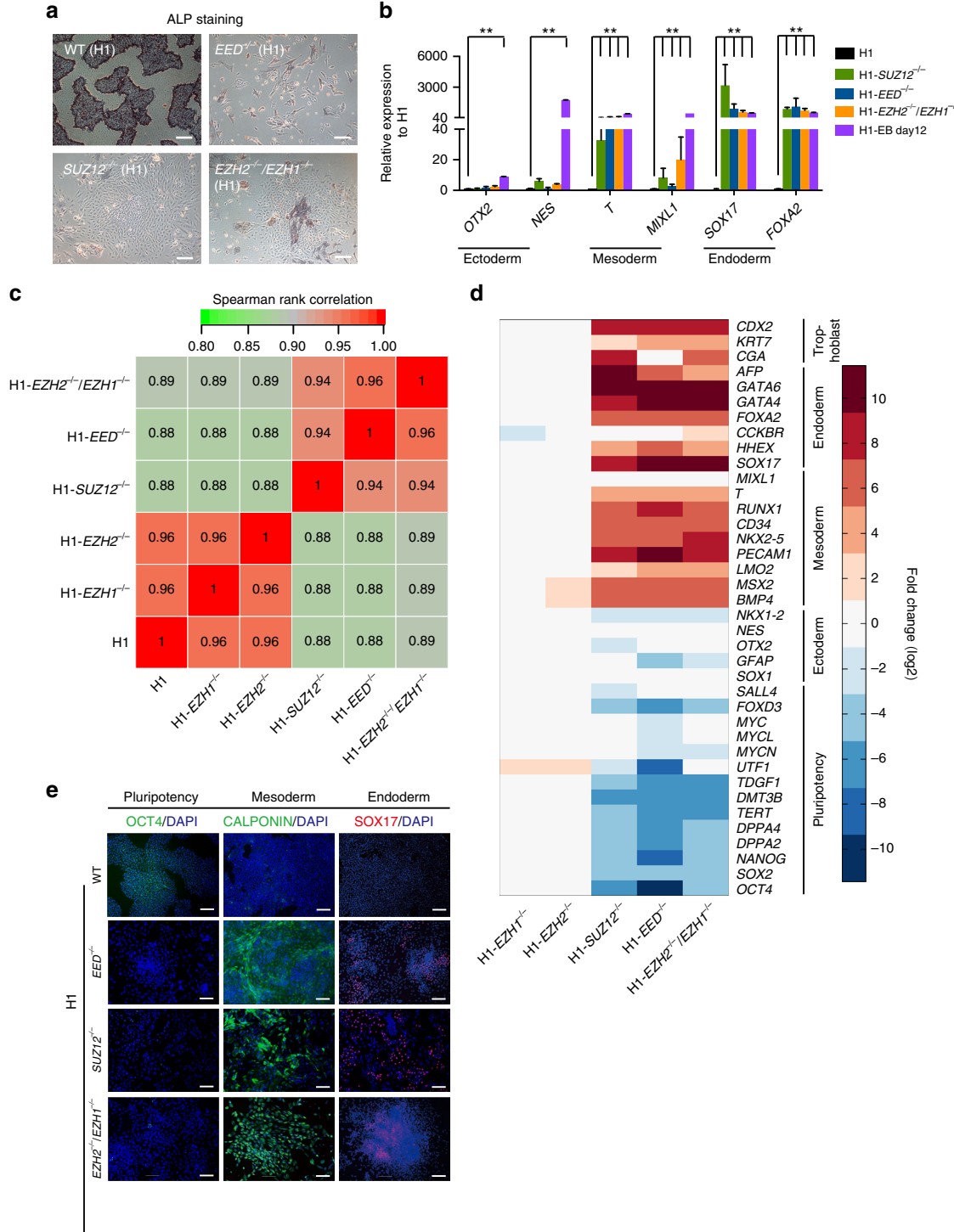

**Fig. 2** PRC2$^{-/-}$ hESCs exhibit spontaneous differentiation to meso-endoderm fate. **a** Morphology and alkaline phosphatase (ALP) activity staining on each indicated hESCs. *Scale bar*, 200 μm. **b** qRT-PCR analysis on the selected lineage genes in the indicated cell lines. Negative control: H1, positive control: H1 cells-derived embryonic bodies (H1-EB day 12). Significance level was determined using unpaired two-tailed Student's *t* tests. \*\*, *P* < 0.01. The data represent mean ± SD from three independent repeats. **c** Spearman rank correlation analysis on the whole-genome transcriptome of indicated cell lines. **d** Heatmap on the selected pluripotent and linage marker genes in the indicated hESCs. We set the expression level of genes in H1 hESCs as 1 and calculated the fold change (log2) of individual gene in none of core component of PRC2 in H1 hESCs, respectively. **e** Immunostaining on the pluripotency and lineage markers, OCT4 (pluripotency), CALPONIN (mesoderm), SOX17 (endoderm) in the indicated cell lines. *Scale bar*, 100 μm. See also Supplementary Fig. 2

***EZH1* and *EZH2* specify early neural ectoderm fate**. hESCs with single deletion of *EZH1* or *EZH2* stayed in an undifferentiated state but had decreased levels of H3K27me3 modifications (Fig. 1c, e). Therefore, $EZH1^{-/-}$ or $EZH2^{-/-}$ H1 hESCs provide a good model to examine the role of PRC2 components in later lineage specifications. We first showed that $EZH1^{-/-}$ or $EZH2^{-/-}$ H1 hESCs cultured under defined conditions that support hPSCs retain typical pluripotency characteristics, as demonstrated by

ALP staining, expression of well-known hESC markers, typical cell-cycle distribution for hESCs and low expression levels of differentiation markers (Fig. 3a–d, Supplementary Fig. 3a). To examine the differentiation potential of these cells, we injected $EZH1^{-/-}$ or $EZH2^{-/-}$ H1 hESCs into immuno-deficient mice and monitored teratoma formation. We observed normal teratoma formation for $EZH1^{-/-}$ or $EZH2^{-/-}$ H1 hESCs in all 3 injected mice. However, on the basis of H&E staining of the teratoma

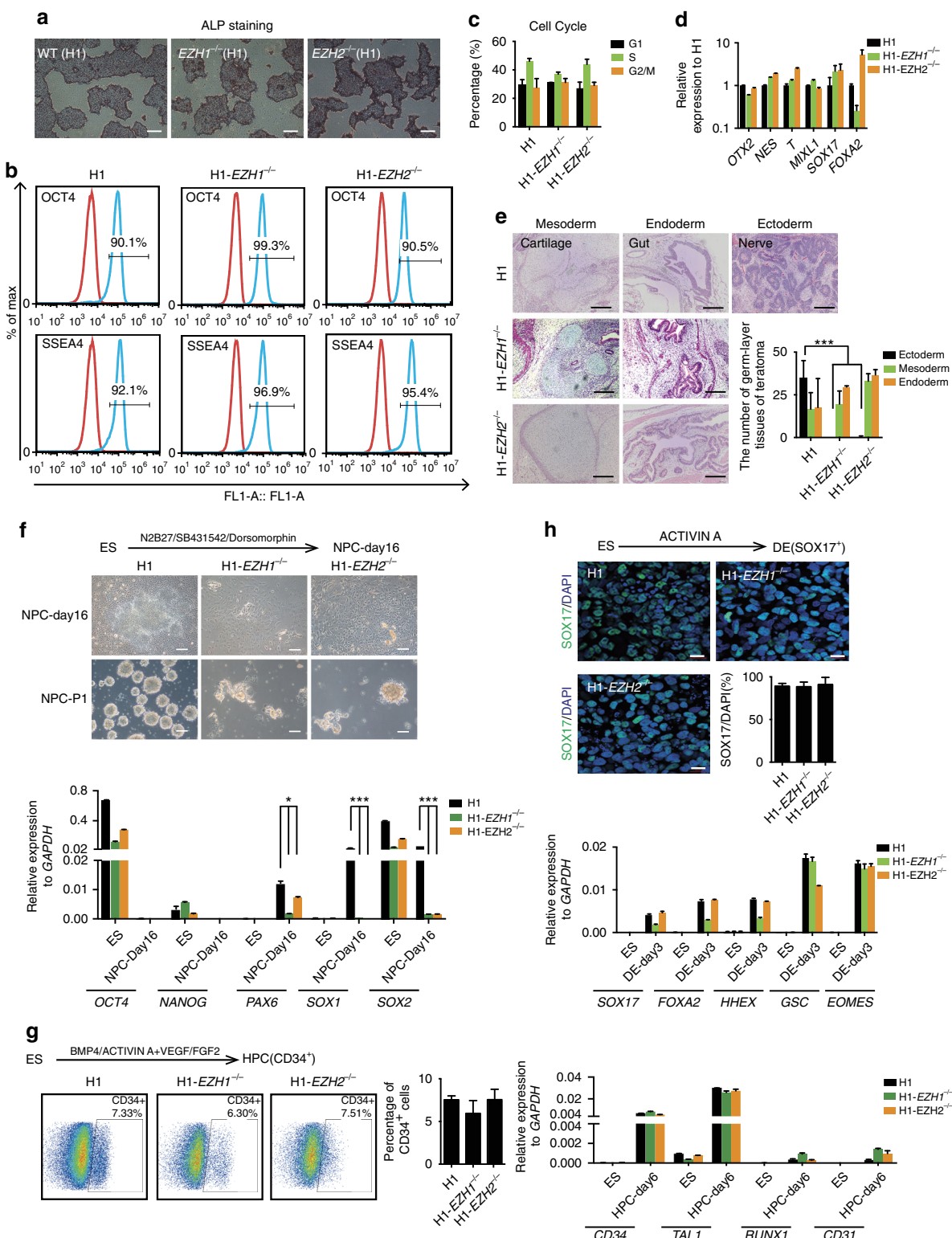

sections, we did not observe any ectoderm tissue from teratomas formed from $EZH1^{-/-}$ or $EZH2^{-/-}$ H1 cells (Fig. 3e, Supplementary Fig. 3b). In contrast, typical mesoderm and endoderm tissues were readily detected in $EZH1^{-/-}$ or $EZH2^{-/-}$ H1 teratomas and were similar to those in WT H1 hESC teratomas (Fig. 3e, Supplementary Fig. 3b). These data indicated that hESCs lacking $EZH1$ or $EZH2$ completely fail to specify neural ectoderm lineages *in vivo*. We then performed directed in vitro differentiation of $EZH1^{-/-}$ or $EZH2^{-/-}$ H1 cells into specific lineages representing the three embryonic germ layers. For neural ectoderm, we induced neural differentiation of hESCs through a well-established and efficient protocol based on dual-SMAD inhibition[40]. Through dual SMAD inhibition, WT H1 hESCs were efficiently converted into neural progenitor cells (NPCs) expressing known NPC markers and forming typical neural spheres in suspension (Fig. 3f). In contrast, $EZH1^{-/-}$ or $EZH2^{-/-}$ H1 hESCs differentiated into only epithelial-like cells without expressing any neural lineage markers and forming neural spheres (Fig. 3f). For mesoderm, we induced differentiation of blood cells by using a published protocol with stromal cell-free condition[42]. Similar percentages of CD34+ hematopoietic progenitor cells were obtained from the differentiation of both WT and $EZH1^{-/-}$ or $EZH2^{-/-}$ H1 hESCs through cytokine treatment (Fig. 3g)[42]. In addition, the known early and late mesoderm or blood lineage marker genes were also successfully induced in $EZH1^{-/-}$ or $EZH2^{-/-}$ H1 hESCs (Fig. 3g). Then, we demonstrated very similar endoderm differentiation between WT and $EZH1^{-/-}$ or $EZH2^{-/-}$ H1 hESCs, on the basis of the expression of SOX17 and several other endoderm markers (Fig. 3h), by using a previously published protocol[43, 44]. Together, our data demonstrate that $EZH1$ and $EZH2$ are required to specify the neural ectoderm lineage in hESCs but is dispensable for mesoderm or endoderm lineage.

**PRC2 deletion preferentially induces BMP signalling in hESCs.** Because the disruption of PRC2 in hESCs resulted in spontaneous differentiation, we then sought to investigate the underlying molecular events that drive differentiation. Because hESCs with PRC2 disruption ($EED^{-/-}$ H1, $SUZ12^{-/-}$ H1 or $EZH2^{-/-}/EZH1^{-/-}$ H1 hESCs) undergo differentiation and cannot be maintained in vitro, we designed an inducible system to rescue the self-renewal capacity of hESCs with PRC2 disruption (Fig. 4a, b). We first introduced an inducible system to over-express (OE) $EED$ in hESCs and subsequently performed gene targeting to knockout the endogenous $EED$ in these cells (Fig. 4a, b, Supplementary Fig. 4a) (see "Methods" section)[45]. This hESC line is referred to as H1-$EED^{-/-}/EED$-OE. The EED level and PRC2 are maintained by DOX treatment during regular cell passaging (Fig. 4d). In the presence of DOX, H1-$EED^{-/-}/EED$-OE maintained an undifferentiated state and could be normally passaged under defined

conditions that support hPSCs (Fig. 4c). However, they underwent gradual and spontaneous differentiation after withdrawal of DOX, despite being kept in hESC medium containing high concentrations of FGF2 to support self-renewal[46, 47]. After DOX withdrawal, H1-$EED^{-/-}/EED$-OE began to morphologically differentiate at day 16 and became fully differentiated at day 20 (Fig. 4c). Moreover, the pluripotency marker genes such as $OCT4$, $SOX2$ and $NANOG$ were fully repressed at later stages of DOX withdrawal (Fig. 4e). In agreement with the data shown in Fig. 2, the meso-endoderm genes but not neural ectoderm genes were activated at later stages of DOX withdrawal (Fig. 4f). We then performed whole-genome transcriptome analysis on H1-$EED^{-/-}/EED$-OE at different time points after DOX withdrawal. Spearman's rank correlation analysis on the whole transcriptome clearly showed a gradual differentiation of H1-$EED^{-/-}/EED$-OE after DOX withdrawal (Fig. 4g). Again, on the basis of the transcriptome data, meso-endoderm genes, but not ectoderm genes, were gradually activated during the time course of DOX withdrawal (Fig. 4h). The differentiation of H1-$EED^{-/-}/EED$-OE into the meso-endoderm fate was further confirmed by immunostaining for marker genes specific for the three embryonic germ layers (Supplementary Fig. 4b). In searching for pathways that were responsible for the differentiation caused by PRC2 disruption, we identified that TGF-β/BMP signalling factors were clearly up-regulated at very early stages of DOX withdrawal (Fig. 4i, Supplementary Fig. 4c–f). BMPs and related factors such as $BMP2$, $BMP4$, $BMP7$, $GDF6$ and $ID2$ began to increase early at day 8 even when no obvious differentiation had been detected on the basis of morphology and the transcriptome data (Fig. 4c, g). Together, these data demonstrated that disruption of PRC2 in hESCs preferentially induces BMP signaling at early stage.

**Inhibition of BMP signalling rescues PRC2 deficiency in hESCs.** We then sought to examine whether the early induction of BMP signalling after PRC2 disruption might be the major reason for the differentiation of hESCs (Fig. 5a). To test this hypothesis, in addition to H1-$EED^{-/-}/EED$-OE, we also prepared additional inducible systems to rescue the functional loss of other PRC2 components, for example, $EZH1$ or $EZH2$. Similarly, we first introduced the inducible system to over-express $EZH1$ or $EZH2$ in $EZH2^{-/-}$ or $EZH1^{-/-}$ H1 hESCs and then deleted endogenous $EZH1$ or $EZH2$ in the same cell lines (Fig. 5a, Supplementary Fig. 4a). These cells, denoted H1-$EZH2^{-/-}/EZH1^{-/-}/EZH2$-OE or H1-$EZH2^{-/-}/EZH1^{-/-}/EZH1$-OE, maintained a normal undifferentiated state in the presence of DOX (Fig. 5b, Supplementary Fig. 5a–d). As expected, after withdrawal of DOX, H1-$EZH2^{-/-}/EZH1^{-/-}/EZH2$-OE, H1-$EZH2^{-/-}/EZH1^{-/-}/EZH1$-OE and H1-$EED^{-/-}/EED$-OE all displayed complete differentiation at 20 or more days of culture (Fig. 5b). However, the differentiation was blocked by treatment with dorsomorphin

**Fig. 3** $EZH1$ and $EZH2$ specify early neural ectoderm fate. **a** Morphology and alkaline phosphatase ($ALP$) activity staining on each indicated hESCs. *Scale bar*, 200 μm. **b** FACS analysis on the expression of indicated pluripotent markers in the indicated hESCs. **c** Cell cycle of the indicated hES cell lines. The data represent mean ± SD from three biological repeats. **d** qRT-PCR analysis on lineage genes in the indicated hESCs. Wild-type H1 hESCs serve as control. The data represent mean ± SD from three biological repeats. **e** H&E staining on sections of teratomas formed by the indicated hESC cell lines. *Scale bar*, 200 μm. Significance level was determined using unpaired two-tailed Student's *t* tests. ***, $P < 0.001$. The data represent mean ± SD from three biological repeats. **f** Neural differentiation of the indicated hESC cell lines. hESCs were treated by SB431542/Dorsomorphin (DM) (5/5 μM) in N2B27 medium in monolayer condition and analysed by qRT-PCR on lineage markers and neural sphere formation. *Scale bar*, 100 μm. Significance level was determined using unpaired two-tailed Student's *t* tests. *, $P < 0.05$. ***, $P < 0.001$. The data represent mean ± SD from three independent repeats. **g** Blood differentiation. The indicated hESCs were treated in the cocktail of indicated cytokines in monolayer condition and analyzed by qRT-PCR on indicated blood lineage markers or FACS analysis on CD34+ cells. The data represent mean ± SD from three independent repeats. **h** Endoderm differentiation. The indicated hESCs were treated by activin A (100 ng mL$^{-1}$) in defined condition. The endoderm marker, SOX17 was analyzed by immunostaining and other endoderm lineage markers were analyzed by qRT-PCR. *Scale bar*, 20 μm. The data represent mean ± SD from three independent repeats. DE, definitive endoderm cell; ES embryonic stem cell; HPC, hematopoietic progenitor cell; NPC, neural progenitor cell. See also Supplementary Fig. 3

(DM), a BMP inhibitor (Fig. 5b)[48]. As a control, another compound, SB431542, that specifically inhibits TGF-β/Nodal signalling had no effect on the differentiation process (Fig. 5b)[40]. Moreover, other reported BMP inhibitors such as DMH1[49] and LDN193189[49, 50] also completely blocked differentiation after DOX withdrawal in H1-*EZH2*$^{-/-}$/*EZH1*$^{-/-}$/*EZH2*-OE (Supplementary Fig. 6d). Further, using qRT-PCR, immunostaining and

ALP staining (Fig. 5c, d, and Supplementary Fig. 6a–f), we confirmed that whereas OCT4 was down-regulated and the meso-endoderm genes SOX17 and CALPONIN were upregulated after DOX withdrawal, BMP inhibitor treatment reversed these phenotypes in all three examined hESC lines (Fig. 5c, d). These data together demonstrate that PRC2 regulates the loss of pluripotency by suppressing BMP signalling.

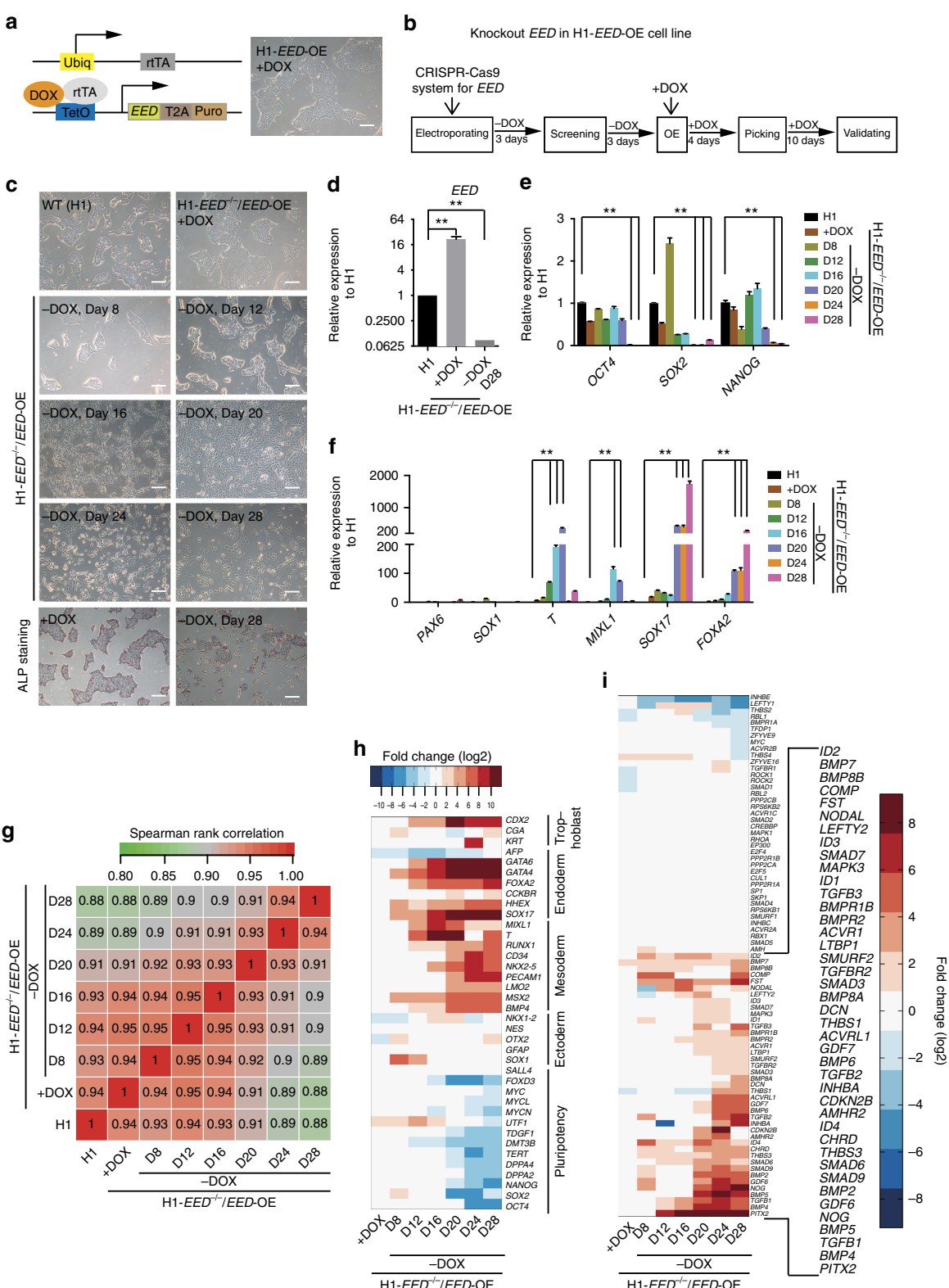

**PRC2 is required for pluripotency in primed not naive state**. The apparent discrepancy between our results and those reported in mESCs may reflect a differential requirement of PRC2 in naive and primed states[51–55]. To test this possibility, we generated PRC2 component gene knockout cell lines in conical OG2 mESCs with GFP controlled by the Oct4 promoter (Oct4: GFP)[56]. Two critical PRC2 genes, *Suz12* and *Eed*, were selected for gene targeting in mESCs (Fig. 6a, Supplementary Fig. 7a–e). As expected, and in agreement with previous reports, mESCs deficient in *Suz12* (mESCs-*Suz12*[−/−]) or *Eed* (mESCs-*Eed*[−/−]) exhibited a normal phenotype and expressed Oct4 when they were maintained under typical conditions for mESC growth (see Methods) (Fig. 6a, Supplementary Fig. 7d) despite the upregulation of *Bmp4* (Fig. 6c). However, when converted into the primed state through a well-established protocol[52], mESCs-*Suz12*[−/−] and mESCs-*Eed*[−/−] but not WT mESCs displayed a gradual and spontaneous differentiation, as indicated by morphology changes and a loss of Oct4: GFP (Fig. 6a, b). The differentiation phenotype was further confirmed by the downregulation of pluripotency marker genes and upregulation of differentiation genes and TGF-β/BMPs signalling factors in mESCs-*Suz12*[−/−] and mESCs-*Eed*[−/−] (Fig. 6c, Supplementary Fig. 7f). These results suggested that PRC2 is dispensable for maintaining pluripotency in naive but is required for primed mouse ESCs.

Additionally, hESCs are regarded as being in a primed state that can be converted to a naive state by over-expressing *NANOG* and *KLF2* or switching to a medium containing specific growth factors and small molecules[57, 58]. To test whether PRC2 functions similarly in naive and primed states in human cells, we first knocked down *EZH1* by shRNA in H1-*EZH2*[−/−] hESCs, thus resulting in an expected differentiation when the cells were in a primed state (Fig. 6d, e, Supplementary Fig. 6g, h). However, when the cells were converted to a naive state by over-expression of *NANOG/KLF2*, H1-*EZH2*[−/−] hESCs became resistant to differentiation after *EZH1* knockdown (Fig. 6d, e, Supplementary Fig. 6g, h)[57]. Furthermore, pluripotency marker genes such as *OCT4*, *SOX2*, *NANOG* were well maintained in the converted naive state of H1-*EZH2*[−/−] hESCs, whereas the same genes were significantly downregulated in H1-*EZH2*[−/−] hESCs after *EHZ1* knockdown in an "unconverted" primed state (Fig. 6e, Supplementary Fig. 6h). These data suggested that in human pluripotent stem cells, PRC2 is also required for pluripotency in the primed state and not the naive state. To further confirm whether this phenomenon is a general mechanism of PRC2 function in naive pluripotency, we also examined other reported protocols to obtain hESCs in a naive state[58]. As reported[58], we converted WT H1 cells, H1-*EED*[−/−]/*EED*-OE or H1-*SUZ12*[−/−]/*SUZ12*-OE cells into naive state by switching them to medium containing hLIF/ACTIVIN A plus 5 small molecules (5i/L/A), as demonstrated by the up-regulation of marker genes of naive pluripotency (Fig. 6f, Supplementary Fig. 6i). In agreement with the data above, after DOX withdrawal, H1-*EED*[−/−]/*EED*-OE or H1-*SUZ12*[−/−]/*SUZ12*-OE maintained in a primed state underwent differentiation (Fig. 6f, Supplementary Fig. 6i). In contrast, H1-*EED*[−/−]/*EED*-OE or H1-*SUZ12*[−/−]/*SUZ12*-OE in naive state maintained their undifferentiated phenotype, as indicated by morphology and marker gene expression (Fig. 6f, Supplementary Fig. 6i). Together, our data demonstrate that PRC2 is required for pluripotency in primed but not naive ESCs.

## Discussion

PRC complexes are essential regulators of cell lineage decisions during development in different species, such as *Drosophila* and mice[10, 24–26, 59]. At the molecular level, PRCs repress gene expression by mediating histone modifications such as H3K27me3 deposition and through other epigenetic mechanisms[11, 14, 33]. Mutations in each individual core component such as *Eed*, *Suz12* and *Ezh2* in mice result in early lethality due to gastrulation defects[23, 25, 27]. However, the molecular mechanisms underlying the gastrulation defects caused by PRC2 deficiency have not been fully elucidated. In this report, we showed that PRC2-deficient hESCs undergo spontaneous differentiation toward the meso-endoderm germ layers without neural ectoderm, thus suggesting that PRC2 is required for the specification of the ectoderm fate at early stages of differentiation. Furthermore, our data demonstrated that PRC2 is required for maintaining pluripotency in a primed state but is dispensable in the naive state. Therefore, our analyses provide more insights into the understanding of cell fate decisions mediated by PRC complexes during lineage specification at very early stages of development (Fig. 6g).

PRC2 is part of a large network of chromatin regulators that cooperatively govern the genomic architecture. In this work, we sought to begin to understand their exact roles in the dynamic regulation of cell fate decisions. For example, the PRC2-driven H3K27me3 landscape together with H3K4me3 is considered to be critical in repressing lineage genes in a "poised" state that can respond quickly to the differentiation stimuli for lineage commitment[16, 17, 60, 61]. However, our data revealed a specific role of PRC2 in lineage specification as well as their differential requirement in maintaining naive or primed state of pluripotency. Thus, it is not clear how the epigenetic state such as the "poised state" plays a critical role during cell fate decisions into specific lineages. To answer that question, a similar approach as ours could be used to investigate the mediators of H3K4me3 deposition. Given the preliminary data on the role of PRC2 in

**Fig. 4** PRC2 deletion preferentially induces BMP signalling in hESCs. **a** Diagram of lentiviral-based inducible system for *EED* expression. *EED* expression was controlled by DOX treatment. The morphology of H1 hESCs with *EED* over-expression is shown. *Scale bar*, 200 μm. **b** Strategy of *EED* knockout in H1 hESCs with *EED* over-expression. CRISPR/Cas9 and targeting vector were transfected into H1 hESCs with DOX inducible *EED* overexpression (H1-*EED*-OE). The positive cell clones were then selected by puromycin in the absence of DOX for the first 10 days and expanded in the presence of DOX). The positive clones were then isolated and maintained in defined medium with DOX. The correctly targeted cells, referred as H1-*EED*[−/−]/*EED*-OE were confirmed by genomic PCR and qRT-PCR on endogenous *EED*. **c** Gradual differentiation of H1-*EED*[−/−]/*EED*-OE upon withdrawal of DOX, indicated by morphology and ALP staining. *Scale bar*, 200 μm. **d** qRT-PCR examination on the expressions of *EED* in H1-*EED*[−/−]/*EED*-OE at 28 days after DOX withdrawal. Significance level was determined using unpaired two-tailed Student's *t* tests. **, *P* < 0.01. The data represent mean ± SD from three independent repeats. **e, f** qRT-PCR examination on the expressions of indicated pluripotent and lineage genes in H1-*EED*[−/−]/*EED*-OE at different time points after DOX withdrawal. Significance level was determined using unpaired two-tailed Student's *t* tests. **, *P* < 0.01. The data represent mean ± SD from three independent repeats. **g** Spearman's rank correlation analysis on the whole-genome transcriptome of H1-*EED*[−/−]/*EED*-OE at different time points after DOX withdrawal. **h** Heatmap on the selected pluripotent and lineage marker genes in H1-*EED*[−/−]/*EED*-OE at different time points after DOX withdrawal. We set the expression level of genes in H1 hESCs as 1 and calculated the fold change (log2) of individual gene in H1-*EED*[−/−]/*EED*-OE hESCs, respectively. **i** Heatmap on TGF-β/BMPs signaling genes in H1-*EED*[−/−]/*EED*-OE at different time points after DOX withdrawal. We set the expression level of genes in H1 hESCs as 1 and calculated the fold change (log2) of individual gene in H1-*EED*[−/−]/*EED*-OE hESCs, respectively. See also Supplementary Fig. 4

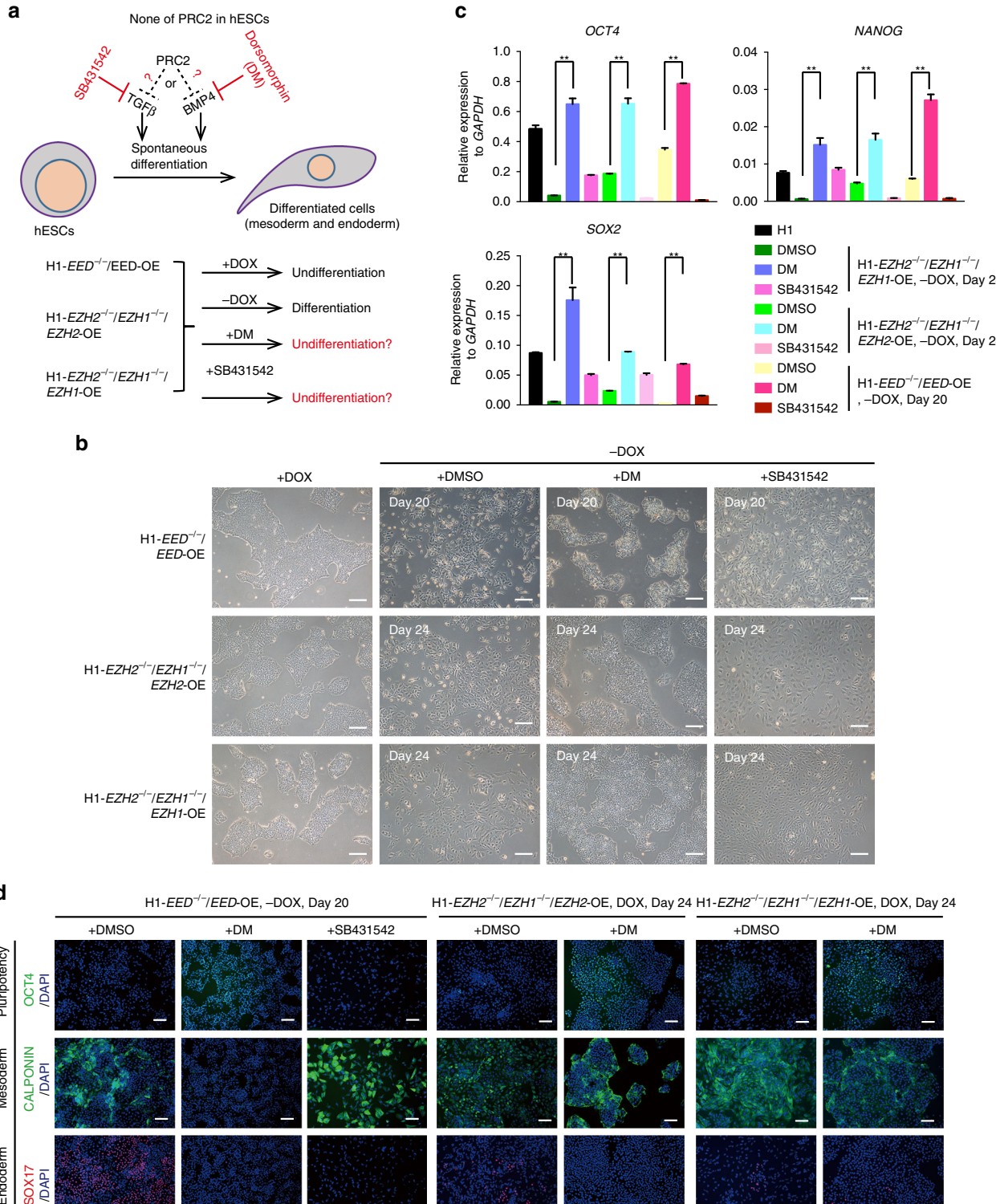

**Fig. 5** Inhibition of BMP signalling rescues PRC2 deficiency in hESCs. **a** Strategy of PRC2 disruption rescue experiments. H1-*EZH2*^−/−^/*EZH1*^−/−^/*EZH2*-OE or H1-*EZH2*^−/−^/*EZH1*^−/−^/*EZH1*-OE was prepared as described in Fig. 4a, b. H1-*EED*^−/−^/*EED*-OE, H1-*EZH2*^−/−^/*EZH1*^−/−^/*EZH2*-OE and H1-*EZH2*^−/−^/*EZH1*^−/−^/*EZH1*-OE were cultured in defined medium in the absence of DOX, but with adding with BMP or TGF-β inhibitors (1 μM DM or 5 μM SB431542) for 20 more days. **b** Morphology of the indicated hESCs cultured in defined medium with indicated condition. DM while not SB431542 treatment rescued morphological change triggered by DOX withdrawal. *Scale bar*, 200 μm. **c** Expression of pluripotent genes *OCT4*, *SOX2* and *NANOG* in the indicated hESCs with different treatments. Significance level was determined using unpaired two-tailed Student's *t* tests. \*\*, *P* < 0.01. The data represent mean ± SD from three independent repeats. **d** Immunostaining on the pluripotency and lineage markers, OCT4 (pluripotency), CALPONIN (mesoderm), SOX17 (endoderm) in the indicated hESCs with different treatments. *Scale bar*, 100 μm. See also Supplementary Figs. 5 and 6

lineage and cell fate determination, we were particularly surprised by the degree of its specificity toward the ectoderm lineage through the suppression of the meso-endoderm. This phenotype indicates the possibility that signalling pathways and developmental factors for the meso-endoderm lineage are preferentially de-repressed after PRC2 disruption. Indeed, we showed that BMP signalling plays a critical role in this process. Mechanistically, more work is needed to further delineate PRC2 and BMP pathways.

H3K27me3 modifications were completely abolished after deletion of PRC2 components. These data indicated that the roles of PRC2 and H3K27me3 in regulating lineage specification is specific and not as broad as previously thought[23, 31, 34, 62]. Interestingly, hESCs with single deletion of *EZH1* or *EZH2* maintained a certain level of H3K27me3 and were in a typical undifferentiated state, in agreement with functional redundancy between EZH1 and EZH2 in catalyzing histone modifications[7, 39]. However, these cells were completely defective in the generation

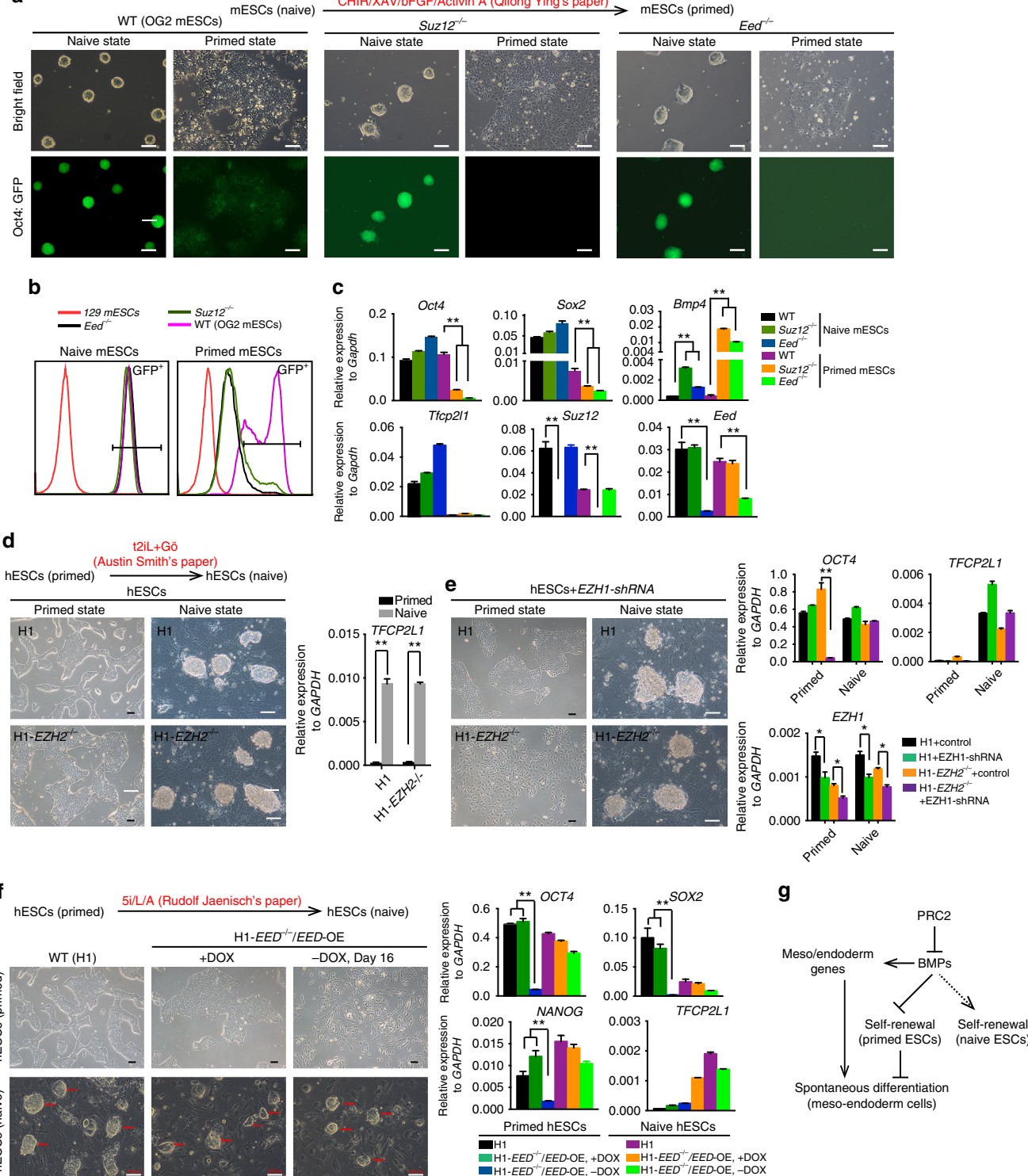

of neural ectoderm lineages, but the specifications of other lineages remained unaffected (Fig. 3). The detailed molecular mechanisms underlying neural ectoderm defect in H1-$EZH1^{-/-}$ or H1-$EZH2^{-/-}$ cells remain unknown. One possibility is that the decreased level of H3K27me3 in H1-$EZH1^{-/-}$ or H1-$EZH2^{-/-}$ failed to fully repress BMPs that might dominantly switch the cell fate to meso-endoderm lineages during differentiation. The detailed molecular mechanisms underlying neural ectoderm defect in H1-$EZH1^{-/-}$ or H1-$EZH2^{-/-}$ cells require further investigation.

We initially encountered an intriguing dichotomy for PRC2 disruption in hESCs and mESCs. We observed a clear differentiation phenotype in hESCs with deletion of each core component of PRC2, but mESCs-$Suz12^{-/-}$ or mESCs-$Eed^{-/-}$ maintained a relatively normal undifferentiated phenotype. hESCs and mESCs have been considered to represent different states of pluripotency, primed vs. naive[53–55]. Interestingly, after conversion into a primed state, mESCs-$Suz12^{-/-}$ or mESCs-$Eed^{-/-}$ clearly exhibited spontaneous differentiation, as observed in hESCs (Fig. 6)[52]. Moreover, disruption of PRC2 in "naive" hESCs that were generated in vitro resulted in substantially less impairment of pluripotency (Fig. 6)[57, 58]. Therefore, PRC2 is differentially required for maintaining pluripotency for cells in different states, i.e., it indispensable for the primed state but unnecessary for the naive state. The molecular mechanisms of how PRC complexes maintain the cellular identity of naive and primed PSCs remain to be fully elucidated. Nonetheless, the differential requirement of PRC2 may serve as a molecular signature for distinguishing naive vs. primed states in hESCs.

## Methods

**Cell culture.** Human ESC lines H1 (Wi Cell), H9 (Wi Cell) and knockout cell lines were maintained in mTeSR1 (STEMCELL Technologies) on matrigel (Corning)-coated plates. Mouse ESC cell line OG2 with GFP controlled by Oct4 promoter, was kindly provided by Dr. Jiekai Chen. OG2 mESCs and knockout mESCs based on OG2 mESCs were maintained on feeder layers in mESC + 2iL medium (DMEM/ high glucose (Hyclone), 15% FBS (Gibco), NEAA (Gibco, 100×), GlutaMAX (Gibco, 100×), Sodium Pyruvate (Gibco, 100×), 1 μM PD0325901 (Selleck), 3 μM CHIR99021 (Selleck), 1000 units/mL mLIF). OG2 mESCs and knockout mESCs based on OG2 mESCs were maintained on gelatin (Millipore)-coated plate in mouse N2B27 + 2iL medium (50% DMEM/High glucose (Hyclone), 50% Knockout DMEM (Gibco), N2 (Gibco, 200×) + B27 (Gibco, 100×), NEAA (Gibco, 100×), GlutaMAX (Gibco, 100×), Sodium Pyruvate (Gibco, 100×), 1 μM PD0325901 (Selleck), 3 μM CHIR99021 (Selleck), 100 μM β-mercaptoethanol (Gibco), 1000 units mL$^{-1}$ mLIF). All cell types were maintained at 5% $CO_2$.

**Gene knockout in human and mouse ESCs.** pX330 (Addgene) can express Cas9 protein and guide RNA. Guide RNAs (gRNAs) for *EZH1*, *EZH2*, *EED*, *SUZ12*, mouse *Suz12* and *Eed* were designed on the website (crispr.mit.edu)[37]. Donor DNAs of these genes containing left and right homology arms, a LoxP-flanked PGK-puromycin cassette or a LoxP-flanked PGK-neomycin cassette, and they were

used for targeting. For targeting, $1 \times 10^6$ hESCs were electroporated with 2 μg of donor DNA and 4 μg of pX330 plasmid containing gRNA for each gene respectively. Then the electroporated hES cells were plated onto matrigel-coated six-well plates with Y-27632 (10 μM, Sigma) for 1 day. Positive clones were selected by puromycin (1 μg mL$^{-1}$, Gibco) or G418 (100 μg mL$^{-1}$, Sigma) in mTeSR1. To get double deletion of *EZH1* and *EZH2*, the antibiotic cassettes in H1-$EZH1^{-/-}$ or H1-$EZH2^{-/-}$ were deleted using Cre-LoxP system. $1 \times 10^6$ hESCs (H1-$EZH1^{-/-}$ or H1-$EZH2^{-/-}$) were electroporated with 400 ng of Cre mRNA, respectively. Then 500 electroporated hES cells were plated onto matrigel-coated six-well plates with Y-27632 (10 μM, Sigma) for 2 days. After deletion of antibiotic cassette in H1-$EZH1^{-/-}$ or H1-$EZH2^{-/-}$, the protocol of knockout *EZH2* based on H1-$EZH1^{-/-}$ or knockout *EZH1* based on H1-$EZH2^{-/-}$ followed previous procedure. For deletion of mouse *Suz12* or *Eed*, $1 \times 10^6$ mESCs were electroporated with 2 μg of donor DNA and 4 μg of pX330 plasmid containinng gRNA for *Suz12* or *Eed*. Then the electroporated mES cells were plated onto feeder in mESC + 2iL medium. Positive clones were selected by puromycin (1 μg mL$^{-1}$, Gibco) or G418 (100 μg mL$^{-1}$, Sigma) in mESC + 2iL medium. All guide RNA sequences and primer sequences are listed in Supplementary Table 1.

**Inducible system for gene knockout in hESCs.** In human ES cell line (H1), we firstly induced an inducible over-expression (OE) system[45] for over-expressing *EED* or *SUZ12* respectively, hereafter referred as H1-EED-OE, H1-SUZ12-OE. In human knockout cell line H1-$EZH2^{-/-}$ or H1-$EZH1^{-/-}$, we induced an inducible over-expression (OE) system for over-expressing *EZH2* or *EZH1* respectively, hereafter referred as H1-$EZH2^{-/-}$/EZH2-OE or H1-$EZH1^{-/-}$/EZH1-OE. These cells were selected with 2 μg mL$^{-1}$ doxycycline (DOX) and puromycin. And then endogenous core components of PRC2 were deleted in these inducible OE cell lines respectively, For targeting, $1 \times 10^6$ hESCs were electroporated with 2 μg of donor DNA and 4 μg of pX330 plasmid containing gRNA for each gene respectively. Then the electroporated hES cells were plated onto matrigel-coated 6-well plates with Y-27632 (10 μM, Sigma) for 1 day. Positive clones were selected by puromycin (1 μg mL$^{-1}$, Gibco) in mTeSR1. And we selected positive clones using puromycin for 3 days, and then these clones were picked and cultured in mTeSR1 plus 2 μg mL$^{-1}$ doxycycline (DOX) on matrigel-coated plates. All primer sequences are listed in Supplementary Table 2.

**PCR detection of knockout cell clones.** Genomic DNA of knockout cell clones was extracted with TIANamp Genomic DNA Kit (Tiangen) for PCR analysis. 30–50 ng of genomic DNA templates and KOD PLUS (Toyobo) were used in all PCR reactions. Primer set of each gene including F1 and R1 was used to amplify a 2.5–3.2 Kb product of the targeted integration. Primer set of each gene including F2 and R2 was used to amplify a 2.1–2.8 Kb product product to identify whether random integration occurred. All primers sequences are listed in Supplementary Table 1.

**Western blot analysis.** To detect knockout efficiency of these genes and the methylation of H3K27 and H3K4, cells were lysed on ice in RIPA buffer (Beyotime). Whole-cell extracts were resolved by 10% SDS-PAGE to get knockout efficiency of these genes and by 15% SDS-PAGE to get the methylation of H3K27 and H3K4, and then transferred to PVDF membranes (Millipore), and incubated with primary antibodies over-night at 4 °C. Subsequently, the membranes were incubated with HRP-conjugated secondary antibodies. After the membranes were washed in TBST, HRP was detected by ECL (Beyotime) and visualized by SmartChemi Image Analysis System (Sagecreation). The antibodies were used according to the manufacturer's recommendations. All uncropped western blots can be found in Supplementary Figs. 8 and 9. The information for antibodies used is listed in Supplementary Table 3.

---

**Fig. 6** PRC2 is required for maintaining pluripotency in primed not naive state. **a** Conversion of WT, $Suz12^{-/-}$, or $Eed^{-/-}$ naive mESCs into the primed state. WT mESCs is OG2 mESCs with GFP expression controlled by Oct4 promoter (Oct4: GFP). For conversion into the primed state, WT, $Suz12^{-/-}$, or $Eed^{-/-}$ naive mESCs were treated with indicated conditions in the absence of feeder cells[52]. *Scale bar*, 100 μm. **b** FACS analysis on Oct4: GFP in WT, $Suz12^{-/-}$, or $Eed^{-/-}$ mESCs at different state. **c** qRT-PCR analysis on *Oct4*, *Sox2*, *Tfcp2l1*, and *Bmp4* in WT, $Suz12^{-/-}$, or $Eed^{-/-}$ mESCs at different state. Significance level was determined using unpaired two-tailed Student's *t* tests. **, $P < 0.01$. The data represent mean ± SD from three independent repeats. **d** Conversion of WT or H1-$EZH2^{-/-}$ hESCs into the naive state. hESCs with *NANOG/KLF2* expression were further cultured in switched medium with indicated condition[57]. *TFCP2L1*, a marker gene of naive pluripotency, was detected by qRT-PCR. Significance level was determined using unpaired two-tailed Student's *t* tests. **, $P < 0.01$. The data represent mean ± SD from three independent repeats. *Scale bar*, 100 μm. **e** Knockdown of *EHZ1* in primed or naive state WT or H1-$EZH2^{-/-}$ hESCs. *OCT4*, *TFCP2L1*, and *EZH1* were examined by qRT-PCR in the indicated hESCs with *EZH1* knockdown. Significance level was determined using unpaired two-tailed Student's *t* tests. **, $P < 0.01$. *, $P < 0.05$. The data represent mean ± SD from three independent repeats. *Scale bar*, 100 μm. **f** Conversion of WT or H1-$EED^{-/-}$/EED-OE hESCs into the naive state. hESCs were further cultured in switched medium (5i/L/A) with indicated condition in the presence or absence of DOX[58]. *Left*: Morphology of H1, H1-$EED^{-/-}$/EED-OE with DOX or without DOX in primed or naive state. *Right*: *OCT4*, *SOX2*, *NANOG*, and *TFCP2L1* were examined by qRT-PCR in the indicated hESCs. Significance level was determined using unpaired two-tailed Student's *t* tests. **, $P < 0.01$. The data represent mean ± SD from three independent repeats. *Scale bar*, 100 μm. **g** The model for the requirement of PRC2 in primed and naive ESCs. See also Supplementary Fig. 7

**Flow cytometry analysis**. The cells were trypsined for single cells with 0.25% trypsin-EDTA (Gibco) and fixed with 1% paraformaldehyde for about 20 min at room temperature, washed twice with 2% fetal bovine serum (FBS, Natocor) in PBS. And then cells were permeabilized with 90% methanol for 30 min at 4 °C. After washed, cells were incubated with primary antibodies and isotype control antibodies for 30 min at 37 °C. After washed, cells were incubated with secondary antibodies for 30 min at 37 °C. The cells were washed twice and resuspended in 200 μL PBS, and then analyzed with Accuri C6 (BD Biosciences). The information for antibodies is listed in Supplementary Table 3.

**Quantitative real-time PCR**. Total RNA was extracted with Trizol (Invitrogen), and reverse transcribed with oligo dT (Takara) and RT ACE (Toyobo), and then Quantitative real-time PCR qPCR was performed with CFX96 machine (BIO-RAD) and SsoAdvanced SYBR Green Supermix (BIO-RAD) following the manufacturer's recommendations. *GAPDH* was used for quantitative real-time PCR (qRT-PCR) normalization of human sample, and Gapdh was used for qRT-PCR normalization of mouse sample. All the data were measured in three repeats. All primer sequences are listed in Supplementary Table 2.

**Alkaline phosphatase staining**. The cells were plated on matrigel-coated 6-well plates for alkaline phosphatase (ALP) assay. The cells were fixed with 4% paraformaldehyde for 20 min at room temperature. After washed thrice with 1 × TBST, the cells were processed with ALP buffer (Beyotime) for 5 min. And then, the sample were added with BCIP (Beyotime) and NBT (Beyotime) for 15 min. The usage of regents was followed with the manufacturer's recommendations (Beyotime).

**Immuno-staining assay**. Undifferentiated and differentiated ESCs were seeded onto matrigel-coated 24-well plates. Cells were fixed with 4% paraformaldehyde for 20 min at room temperature, washed thrice with PBS for 5 min each time. Cells were permeabilized with 0.3% triton X-100 (sigma) and 10% goat serum in PBS, meanwhile cells were incubated with corresponding primary antibodies overnight at 4 °C. The cells were washed thrice with PBS for 5 min each time. And then, cells were incubated with secondary antibodies for 1 h at room temperature. The cells were washed thrice with PBS for 5 min each time. The cells were stained with DAPI (Sigma) for 5 min at room temperature. Images were captured with Leica DMI6000B microscope (Leica Microsystems, GmbH). The information for antibodies is listed in Supplementary Table 3.

**Teratoma formation and analysis**. The experiments involving animal research for teratomas formation had been reviewed and approved by IACUC at GIBH (NO. 2010012). The size of tumour growth that was acceptable for ethical approval was about 2.5 × 2.5 cm. In our experiments, $EZH1^{-/-}$, $EZH2^{-/-}$ and wild-type H1 hES cells that were cultured on matrigel-coated 6-well plates were digested by Accuatse (Sigma) for 8 min at 37 °C and resuspended in 30% matrigel (Corning) in DMEM/F12 (Hyclone), and then injected subcutaneously into immuno-deficient mice. We used that the age of mice were about 4 weeks, and the sex of mice were both male and female, and strain of mice were NOD-SCID mice. Teratomas were detected after 8 weeks and fixed in 4% paraformaldehyde, and then stained with hematoxylin/eosin (H&E). The size of tumour growth was about 2 × 2 cm.

**Neural progenitor cells differentiation**. To initiate neural differentiation, hESC were plated onto matrigel-coated 12-well plates with 95–100% of cell confluence, and then these cells were cultured in N2B27 medium plus SB431542 and Dorso-morphin (50% DMEM/F12 (Hyclone), 50% Neurobasal (Gibco), N2 (Gibco, 200×), B27 (Gibco, 100×), Glutamax (Gibco, 200×), NEAA (Gibco, 200×), 5 μg mL⁻¹ insulin(Gibco, 200×), 1 μg mL⁻¹ heparin (Sigma), 5 μM SB431542 (Selleck), 5 μM Dorsomorphin(DM, Selleck))[40]. Every 2 days changing fresh culture medium. After 8 days, the cells were passaged on matrigel-coated 6-well plates in N2B27 medium. After 16 days, canonical neural rossettes of wild-type hESCs appeared and these cells were suspended for neural sphere formation.

**Hematopoietic progenitor cells differentiation**. To initiate hematopoietic differentiation[42], the hESCs were passaged with dispase (2 mg mL⁻¹) onto matri-gel-coated 12-well plates. These cells were cultured in E6 medium[46] plus ACTIVIN A and BMP4 (DMEM/F12 (Hyclone), 64 mg L⁻¹ Lascorbic acid (Sigma), NaCl (Sigma, adjusting the osmolarity to 340 mOsm), ITS –G (Gibco, 100×) and 50 ng mL⁻¹ ACTIVIN A (Sino biological, 10429-HNAH-50), 50 ng mL⁻¹ BMP4 (Peprotech, 120-05ET)) for 2 days. And then cells were cultured in E6 medium plus 40 ng mL⁻¹ VEGF (Sino biological, 10008-HNAB-50) and 50 ng mL⁻¹ bFGF (Sino biological, 10014-HNAE-50) for next two days. For next 3 days, these cells were cultured in E6 medium plus 40 ng mL⁻¹ VEGF, 50 ng mL⁻¹ bFGF, 10 μM SB431542 (Selleck). After 7 days, these cells were collected for extracting total RNA and FACS analysis for CD34 expression. CD34 antibody conjugated with PerCP-Cy5.5 were used according to the manufacturer's recommendations.

**Definitive endoderm cells differentiation**. To initiate definitive endoderm cells (DE cells) differentiation[43, 44], hESCs were cultured for 3 days in RPMI1640 (Gibco) /B27 medium (Insulin minus, Gibco) and 100 ng mL⁻¹ Activin A (Peprotech) on matrigel-coated 24-well plate. After 3 days, these cells were collected for extracting total RNA and immunostaining for SOX17 expression.

**Conversion of primed mouse ESCs**. To induce naive mouse ESCs into primed mouse ESCs (mEpiSCs), conversion of WT, $Suz12^{-/-}$, $Eed^{-/-}$mESCs into the primed state according to Qilong Ying's paper[52]. $3 × 10^5$ mESCs were seeded on gelatin-coated 6-well plate and cultured in mouse N2B27 + 2iL medium for day. And then these cells were cultured in switched medium (50% DMEM/F12 (Hyclone), 50% Neurobasal (Gibco), N2 (Gibco, 200×), B27 (Gibco, 100×), Glutamax (Gibco, 100×), NEAA (Gibco, 100×), 100 μM β-mercaptoethanol (gibco), 3 μM CHIR99021 (Selleck), 2 μM XAV939 (Sigma), 10 ng mL⁻¹ Activin A (Peprotech), 10 ng mL⁻¹ FGF2 (R&D systems)) for 5 days. And then WT mESCs maintained typical primed state.

**Conversion of naive hESCs**. To induce primed hESCs into naive hESCs, we used methods that have been reported by Austin Smith and Rudolf Jaenisch[57, 58]. At first, conversion of WT or H1-$EZH2^{-/-}$ hESCs into the naive state according to Austin Smith's paper. NANOG and KLF2 were overexpressed in hESCs though lentiviral-based inducible vectors. hESCs with NANOG/KLF2 expression were further cultured in switched medium (human N2B27 + 2iL medium: 50% DMEM/F12 (Hyclone), 50% Neurobasal (Gibco), N2 (Gibco, 200×), B27 (Gibco, 100×), Glutamax (Gibco, 200×), NEAA (Gibco, 200×), Sodium Pyruvate (Gibco, 100×), 1 μM PD0325901 (Selleck), 3 μM CHIR99021 (Selleck), 100 μM β-mercaptoethanol (gibco), 1000 units mL⁻¹ mLIF) in the presence of DOX induced transgene expression. These cells were passaged for every week, and cultured on feeder in N2B27 + 2iL medium plus DOX. At about 3–4 passages, these cells maintained typical naive state and expressed the naive marker *TFCP2L1*. And then these naive cells were maintained on feeder in N2B27 + 2iL medium plus 2 μM + Gö6983.

Secondly, conversion of WT or H1-$EED^{-/-}$/EED-OE or H1-$SUZ12^{-/-}$/SUZ12-OE hESCs into the naive state according to Rudolf Jaenisch's paper. $2 × 10^5$ WT hESCs were seeded on feeder and cultured in KSR medium (80% DMEM/F12 (Hyclone), 15% FBS (Hyclone), 5% KSR (Gibco), Glutamax (Gibco, 200×), NEAA (Gibco, 200×), 100 μM β-mercaptoethanol (gibco), and 4 ng mL⁻¹ FGF2 (R&D systems), 10 μM Y-27632 (Sigma)) for 4 days. $2 × 10^5$ H1-$EED^{-/-}$/EED-OE or H1-$SUZ12^{-/-}$/SUZ12-OE were seeded on feeder and cultured in KSR medium plus DOX for 4 days. Then these cells were trypsinized for single-cell dissociation and cultured on feeder in 5i/L/A medium (50% DMEM/F12 (Hyclone), 50% Neurobasal (Gibco), N2 (Gibco, 200×), B27 (Gibco, 100×), Glutamax (Gibco, 200×), NEAA (Gibco, 200×), 50 μg mL⁻¹ BSA (Sigma), 100 μM β-mercaptoethanol (gibco), 1 μM PD0325901 (Selleck), 1 μM IM-12 (Enzo), 0.5 μM SB590885 (R&D systems), 1 μM WH-4-023 (A Chemtek), 10 μM Y-27632 (Sigma), 20 ng mL⁻¹ hLIF, 20 ng mL⁻¹ Activin A (Peprotech)) for 10 days. We converted H1-$EED^{-/-}$/EED-OE or H1-$SUZ12^{-/-}$/SUZ12-OE hESCs to naive state in the presence or absence of DOX. And then these cells maintained typical naive state. These cells were passaged for every week, and cultured on feeder in 5i/L/A medium.

**RNA-seq and spearman's rank correlation and heatmap analysis**. After the digestion of cultured cells, WT hESCs and the target cells were collected and lysed with 400 μL Trizol (Invitrogen)[45]. Total RNA was extracted using a Directzol RNA MiniPrep kit (Zymo Research) and sequencing libraries was established using a TruSeq RNA Sample Preparation Kits v2 (48 samples) (Illumina) according to the manufacturer's protocol. The samples were run on an NextSeq system with NextSeq 500 Mid Output Kit v2 (150 cycles).

The number of raw reads mapped to human mRNA reference sequence for GRCh38/hg38 using RSEM (rsem-1.2.4)[63], Bowtie2 (v2.2.5), and normalized with EDAseq (v2.2.0)[64]. Gene expression is expressed as "normalized tag count." Other downstream analyses were performed using glbase[65]. In brief, differential expression between differentiation state and the ESC control was analyzed using the edgeR package[66], where raw read counts per gene were normalized using Trimmed Mean of M-values (TMM). Differentially expressed genes were extracted at the cutoff of false discovery rate (FDR) of <=1% and fold change of > =3. We set the expression level of genes in H1 hESCs as 1 and calculated the fold change (log2) of individual gene in none of core component of PRC2 in H1 hESCs, respectively. These selected pluripotent genes and lineage genes were analyzed for heatmap, and expression level of genes in H1 hESCs were set as 1 and the fold change (log2) of individual gene were calculated in other cell lines, respectively. Spearman's correlation coefficients were then computed with TPM values at log2 scale and all correlation coefficients among samples were represented as a heatmap in R.

**Statistical analyses**. In general, Results were presented as mean ± SD calculated using Microsoft Excel and GraphPad Prism at least three biological repeats. Significance level between samples was determined using unpaired two-tailed Student's *t* tests. *P* value < 0.05 was considered statistically significant in the figures. No samples were excluded for any analysis.

**Data availability**. The RNA-Seq data have been deposited in the Gene Expression Omnibus database under the accession code GSE92625. PCR, qRT-PCR and the RNA-seq data have also been deposited in figshare (http://dx.doi.org/10.6084/m9.figshare.5146789). The authors declare that all the data supporting the findings of this study are available within the article and its supplementary information files or from the corresponding upon reasonable request.

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

## Acknowledgements

We thank the lab members in GIBH for their kindly help. This work was supported by the National Key Research and Development Program of China, Stem Cell and Translational Research (2017YFA0102601); the Frontier and Key Technology Innovation Special Grant from the Department of Science and Technology of Guangdong Province (2014B020225006, 2014B020225002, 2014B050504008, 2015B020228003, 2016B030230002, 2016B030229008); the National Basic Research Program of China, 973 Program of China (2015CB964900, 2014CB965200); the National Natural Science Foundation of China (31371514, 31421004, 31500948); Cooperation Grant of Natural Science Foundation of Guangdong Province (2014A030313801, 2014A030312012, 2015A030310229); the Science and Information Technology of Guangzhou Key Project (201508020258, 201506010092, 2015A030310254); the Youth Innovation Promotion Association of the Chinese Academy of Sciences (2015293); International Science & Technology Cooperation Program of China (2014DFA30180); the "Hundred Talents Program" of Chinese Academy of Science (to X.Z.); Science and Technology Planning Project of Guangdong Province, China (2014B030301058), the Guangdong Province Special Program for Elite Scientists in Science and Technology Innovation (to B.L., 2014TQ01R746; to G.P., 2015TX01R203); Natural Science Foundation of Guangdong Province, China (2016A030313167); Science and Technology Planning Project of Guangdong Province, China (2013B050800004).

## Author contributions

G.P., D.P., and Y.S.: Designed the project and wrote the manuscript. Y.S. and Z.L.: Performed most experiments and result analyses. Q.X.: Performed H1-*EZH2*−/−/*EZH1*−/−/*EZH2*-OE and H1-*EZH2*−/−/*EZH1*−/−/*EZH1*-OE hESCs lines and validation of other PRC2 knockout cell lines. B.W.: Performed mouse *Suz12* and *Eed* knockout in mESCs and the induction of primed mESCs. T.Z.: Performed blood differentiation of human ESCs, W.H.: Performed neural differentiation of human ESCs. Q.Z. and Y.J.: Performed *EZH1* and *SUZ12* knockout in H1. Y.Z., K.H., and Y.L.: Performed induction of naive human ESCs. J.Z. and B.L.: Performed teratoma formation of human ESCs. B.S., C.Z., and K.L.: Performed validation of PRC2 knockout cell lines validation. S.L.: Performed definitive endoderm differentiation of human ESCs. S.T., X.W., Q.C., and X.S.: Performed RNA-seq and bioinformatics analysis. B.L., X.Z., X.S., J.W., H.Y., J.C.: Gave suggestions about experiments and provided some experimental materials. All authors read and approved the final manuscript.

## Additional information

**Competing interests:** The authors declare no competing financial interests.

