## [Peer Review File · Nature Communications]

Reviewers' Comments:

Reviewer #1:

Remarks to the Author:

In this manuscript, the authors identify a shift in the role of the PRC2 complex between naive pluripotent stem cells and primed PSCs, corresponding to the different responses of human embryonic stem cells and mouse embryonic stem cells to the loss of PRC2 function. As previously reported, mouse ESCs (naive) do not require PRC2 for self-renewal or pluripotency but they need it to implement correct differentiation. In contrast, human ESCs, known to correspond to primed pluripotent cells, lose pluripotency in the absence of PRC2 and start differentiating. The important finding reported in this manuscript is that differentiation upon loss of PRC2 is exclusively towards meso-endoderm implying a hierarchy of germ layers and a role of PRC2 in neural ectoderm differentiation. These results clarify earlier reports in the literature and provide valuable new insight on the role of PRC2 in the early steps of differentiation. The experiments are comprehensive and the results are well documented.

The manuscript has numerous problems of syntax, grammar and word usage that in some cases may result in ambiguities and confusion about the underlying meaning. These problems would be easily avoided if the text were edited by an Anglophone.

A few specific comments:

p. 4 line 116, Figure 1c. Cell morphology in these pictures is difficult to see and in any case is not adequate evidence of differentiation. I suggest relying less on such images and more on the activation of specific genes.

p. 6, line 187. As described, the results imply that both EZH1 and EZH2, not one or the other, are required for neural differentiation.

It seems very odd that loss of EZH1 alone, retaining the much more enzymatically active EZH2, is sufficient to block neural differentiation. This strongly suggests that it is not just the production of H3K27me3 that is needed but also some function that is specifically dependent on EZH1. For example, EZH1-containing PRC2 has been reported to bind to nucleosomes, to help EZH2-containing PRC2 bind to chromatin and produce chromatin condensation. In addition, EZH1 has been reported to activate expression of MyoG.

p. 9, line 285. Considerable evidence indicates that H2AK119ub does not repress transcription.

p. 10, line 301. The bivalent state is now considered to be a default state for promoters in ES cells that are not DNA-methylated but not transcriptionally activated. It has long been known that it is not required for pluripotency in mouse ESCs.

In general, I find the figures too crowded and overloaded. There is excessive reliance on images of the cell culture, where the cell morphology is often hard to see and in any case is not conclusive evidence. I suggest focusing on the essentials and, whenever possible, moving cell culture images to the supplementary figures.

Reviewer #2:

Remarks to the Author:

In this study Shan et al. examine the role of different members of the PRC2 complex in regulating human and mouse pluripotent stem cells. The authors use gene knockout and conditional replacement to show that in human cells, loss of the genes in this complex leads to differentiation into the mesendodermal lineages. This effect is in turn related to increased expression of members

of the BMP family and associated signalling molecules. Comparative studies in mouse and human naïve pluripotent cells lead the authors to conclude that the PRC2 genes are required for stem cell maintenance in the primed but not naïve state.

The study is in general well presented and the results provide new insight into the biological actions of these important transcriptional regulators in pluripotent stem cells. The conclusions regarding the activity of the factors in primed versus naïve human stem cells are a bit preliminary in that they are based on limited observations with one type of naïve cell, one cell line, and one PRC2 member. There are a number of specific issues relating to the data and presentation that the authors need to clarify or address.

The manuscript, while clearly written, is in need of editorial attention to minor matters of English grammar and usage.

Specific comments:

1. Page 4 and Figure 1e-it might be best to reverse the panels in 1e so that the figure follows the presentation of the results.
2. Page 4-5: if the knockout cells lack pluripotency factors and differentiate spontaneously how were they maintained for study? At what time point following targeting do the cell lines lose pluripotency factor expression and undergo differentiation?
3. Page 5 Figure 2b-regarding the H1 control, are these cells maintained in the undifferentiated state? It would be important to show differentiated controls as well, for comparison with the knockout lines.
4. Page 5 Figure 1d see comment 3 above
5. Page 5 Figure 2d-how were the representative neural markers selected for display in this figure?
6. Page 6-clarify the point that Figure 3e is making-is the idea to show that these differentiation markers are not elevated in the knockout lines?
7. Page 6 Figure 3f-I appreciate that it is difficult to illustrate something that is not present, but large black crosses in 3f are not very informative. The table is better but should indicate how many teratomas were examined and how many sections of each teratoma were studied.
8. Page 7 figure 4d-it is curious that the pluripotency genes take so long to decay-is it possible that high EED levels in the overexpression context might be affecting this.
9. Figure 4d-it would be better to use color, or different marks within the bars, to distinguish the various timepoints. It is a little difficult to discern the shades of grey.
10. Page 8 Figure 5-the conclusion that BMP signalling is critical is consistent with the data but heavily reliant on the use of one inhibitor, dorsomorphin. This conclusion could be strengthened through the use of noggin or another specific BMP inhibitor.
11. Page 9 Figure 6a-several naïve state marker genes are upregulated in the *Suz12*^{-/-} cells, did the authors explore this further? How do these data relate to the data in 6e?
12. Page 9 Figure 6d-it is difficult to see positive Pou5f1 staining in the WT primed cells, did the authors verify this finding through other means?
13. Page 9 Figure 4f-g: the conclusions here are fairly clear but it is not certain whether all naïve human states would respond the same way or whether other PRC2 members would show similar results.

Reviewers' comments:

Reviewer #1 (Remarks to the Author):

In this manuscript, the authors identify a shift in the role of the PRC2 complex between naive pluripotent stem cells and primed PSCs, corresponding to the different responses of human embryonic stem cells and mouse embryonic stem cells to the loss of PRC2 function. As previously reported, mouse ESCs (naive) do not require PRC2 for self-renewal or pluripotency but they need it to implement correct differentiation. In contrast, human ESCs, known to correspond to primed pluripotent cells, lose pluripotency in the absence of PRC2 and start differentiating. The important finding reported in this manuscript is that differentiation upon loss of PRC2 is exclusively towards meso-endoderm implying a hierarchy of germ layers and a role of PRC2 in neural ectoderm differentiation. These results clarify earlier reports in the literature and provide valuable new insight on the role of PRC2 in the early steps of differentiation. The experiments are comprehensive and the results are well documented.

RE: We thank the reviewer for the positive comment.

The manuscript has numerous problems of syntax, grammar and word usage that in some cases may result in ambiguities and confusion about the underlying meaning. These problems would be easily avoided if the text were edited by an Anglophone.

RE: Thanks for this suggestion. We revised the text.

A few specific comments:

p. 4 line 116, Figure 1c. Cell morphology in these pictures is difficult to see and in any case is not adequate evidence of differentiation. I suggest relying less on such images and more on the activation of specific genes.

RE: Thanks for this suggestion. The reason that the cell images is not clear might due to the reduced resolution in the version for review. Yes, we emphasized the gene activation data included in the main figures and supplementary Figures.

p. 6, line 187. As described, the results imply that both EZH1 and EZH2, not one or the other, are required for neural differentiation.

RE: Thanks for this suggestion. We revised the text accordingly.

It seems very odd that loss of EZH1 alone, retaining the much more enzymatically active EZH2, is sufficient to block neural differentiation. This strongly suggests that it is not just the production of H3K27me3 that is needed but also some function that is specifically dependent on EZH1. For example, EZH1-containing PRC2 has been reported to bind to nucleosomes, to help EZH2-containing PRC2 bind to chromatin and produce chromatin condensation. In addition, EZH1 has been reported to activate expression of MyoG.

RE: Thanks for this suggestion. This is the good point worth to investigate further.

p. 9, line 285. Considerable evidence indicates that H2AK119ub does not repress transcription.

RE: Thanks for this suggestion. We revised the text.

p. 10, line 301. The bivalent state is now considered to be a default state for promoters in ES cells that are not DNA-methylated but not transcriptionally activated. It has long been known that it is not required for pluripotency in mouse ESCs.

RE: Thanks for this suggestion. We revised the text.

In general, I find the figures too crowded and overloaded. There is excessive reliance on images of the cell culture, where the cell morphology is often hard to see and in any case is not conclusive evidence. I suggest focusing on the essentials and, whenever possible, moving cell culture images to the supplementary figures.

RE: Thanks for this suggestion. We revised the Figures and move some of them into the supplementary figures.

--

Reviewer #2 (Remarks to the Author):

In this study Shan et al. examine the role of different members of the PRC2 complex in regulating human and mouse pluripotent stem cells. The authors use gene knockout and conditional replacement to show that in human cells, loss of the genes in this complex leads to differentiation into the mesendodermal lineages. This effect is in turn related to increased expression of members of the BMP family and associated signalling molecules. Comparative studies in mouse and human naïve pluripotent cells lead the authors to conclude that the PRC2 genes are required for stem cell maintenance in the primed but not naïve state.

The study is in general well presented and the results provide new insight into the biological actions of these important transcriptional regulators in pluripotent stem cells. The conclusions regarding the activity of the factors in primed versus naïve human stem cells are a bit preliminary in that they are based on limited observations with one type of naïve cell, one cell line, and one PRC2 member. There are a number of specific issues relating to the data and presentation that the authors need to clarify or address.

RE: Thanks for this suggestion. As suggested, we performed additional experiments and provide new data to strength our conclusion. Besides *Suz12*, we deleted another PRC2 component gene, *Eed* in mouse ESCs. The new data showed that *Eed*^{-/-} mESCs stay in undifferentiated status while at the naïve state, but undergo spontaneous differentiation when converted into the primed state, consistent to the phenotype of *Suz12*^{-/-} mESCs. Then, for human cells, we also examined another protocol published by Rudolf Jaenisch's group to convert hESCs (H1-*EED*^{-/-}/*EED*-OE or H1-*SUZ12*^{-/-}/*SUZ12*-OE) into the naïve state. Again, we found that the naïve state H1-*EED*^{-/-}/*EED*-OE or H1-*SUZ12*^{-/-}/*SUZ12*-OE hESCs converted based on Jaenisch's protocol keep undifferentiated status without PRC2 function upon withdrawal of DOX, while H1-*EED*^{-/-}/*EED*-OE or H1-*SUZ12*^{-/-}/*SUZ12*-OE maintained in primed state were differentiated upon DOX withdrawal. Together, based on the experiments from different PRC2 components, different cell lines and naïve types, our data strongly demonstrate the differential requirement for PRC2 in primed and naïve pluripotent cells. We hope these new data would satisfy. These new data are provided in new Figure 6 and Supplementary Figure 7.

The manuscript, while clearly written, is in need of editorial attention to minor matters of English grammar and usage.

RE: Thanks for this suggestion. We revised the text.

Specific comments:

1. Page 4 and Figure 1e-it might be best to reverse the panels in 1e so that the figure follows the presentation of the results.

RE: Thanks for this suggestion. We revised it. Please see Figure 1e.

2. Page 4-5: if the knockout cells lack pluripotency factors and differentiate spontaneously how were they maintained for study? At what time point following targeting do the cell lines lose pluripotency factor expression and undergo differentiation?

RE: Thanks for this suggestion. The knockout cells exhibited differentiation morphology at 15 days after drug selection, and can be cultured in defined medium (mTeSR1) for another more than 10 days. So, we were able to pick and culture these positive knockout colonies for further gene expression analysis.

3. Page 5 Figure 2b-regarding the H1 control, are these cells maintained in the undifferentiated state? It would be important to show differentiated controls as well, for comparison with the knockout lines.

RE: Thanks for this suggestion. As control, H1 hESCs were cultured in typical mTeSR1 medium to maintain undifferentiated state. We included wild type hESCs derived EBs as positive control for differentiation and revised the figure (Please see Figure 2b).

4. Page 5 Figure 1d see comment 3 above

RE: Thanks for this suggestion. We are not clear which point this comment is about? We guess it's about Figure 2d. In this panel, the transcriptome of undifferentiated H1 cells was served as negative control and the fold change of genes were calculated based on their value in undifferentiated H1 cells. For Figure 2e, we included normal differentiated cells representing three germ layer lineages as positive control for immunostaining (Please see Figure 2b and Supplementary Figure 2e).

5. Page 5 Figure 2d-how were the representative neural markers selected for display in this figure?

RE: Thanks for this suggestion. Those markers were manually selected based on previously published reports (NKX1-2, reference 45. SOX1, NES, OTX2, GFAP, reference 44.).

6. Page 6-clarify the point that Figure 3e is making-is the idea to show that these differentiation markers are not elevated in the knockout lines?

RE: Thanks for this suggestion. Yes, we revised the text to make it clear. Our results showed that these differentiation markers are not elevated in these knockout cell lines and suggest that these cell lines maintain undifferentiated state.

7. Page 6 Figure 3f-I appreciate that it is difficult to illustrate something that is not present, but large black crosses in 3f are not very informative. The table is better but should indicate how many teratomas were examined and how many sections of each teratoma were studied.

RE: Thanks for this suggestion. We revised the Figure and added the quantitative data according to the suggestion. Please see Figure 3e and Supplementary Figure 3b.

8. Page 7 figure 4d-it is curious that the pluripotency genes take so long to decay-is it possible that high EED levels in the overexpression context might be affecting this.

RE: Thanks for this suggestion. EED levels might be the reason, because that EED level went down slowly after withdrawal of DOX and decreased much more significantly at 28 days.

9. Figure 4d-it would be better to use color, or different marks within the bars, to distinguish the various timepoints. It is a little difficult to discern the shades of grey.

RE: Thanks for this suggestion. We revised the Figure. Please see Figure 4d-f.

10. Page 8 Figure 5-the conclusion that BMP signalling is critical is consistent with the data but heavily reliant on the use of one inhibitor, dorsomorphin. This conclusion could be strengthened through the use of noggin or another specific BMP inhibitor.

RE: Thanks for this suggestion. We performed additional experiments and examined other reported BMP inhibitors such as DMH1, LDN193189. These BMPs inhibitors showed similar effects to block differentiation. We had included these new data in Supplementary Figure 6d-f.

11. Page 9 Figure 6a-several naïve state marker genes are upregulated in the *Suz12*^{-/-} cells, did the authors explore this further? How do these data relate to the data in 6e?

RE: Thanks for this suggestion. For the data in Figure 6a, *Suz12*^{-/-} and wild-type mES cells were maintained on feeder cells in mES medium in presence of 2i and LIF. However, in Figure 6e these cells were cultured on gelatin in N2B27 medium with 2i and LIF. The difference might be due to the different culture conditions. To check this hypothesis, we examined the expression level of several naïve marker genes in *Suz12*^{-/-} and *Eed*^{-/-} mES cells maintained in defined condition on gelatin or on feeders. Indeed, we found that these naïve marker genes are higher in these mES cells maintained on gelatin than on feeders. Please see Supplementary Figure 7e. We don't know the reason yet, and the mechanism needs to be investigated further.

12. Page 9 Figure 6d-it is difficult to see positive Pou5f1 staining in the WT primed cells, did the authors verify this finding through other means?

RE: Thanks for this suggestion. We verified Pou5f1 by FACS analysis and qRT-PCR and included new data in Figure 6a-c.

13. Page 9 Figure 4f-g: the conclusions here are fairly clear but it is not certain whether all naïve human states would respond the same way or whether other PRC2 members would show similar results.

RE: Thanks for this suggestion. As suggested, we performed additional experiments and provide new data to strength our conclusion. Besides *Suz12*, we deleted another PRC2 component gene, *Eed* in mouse ESCs. The new data showed that *Eed*^{-/-} mESCs stay in

undifferentiated status while at naïve state, but undergo spontaneous differentiation when converted into the primed state, consistent to the phenotype of *Suz12*^{-/-} mESCs. Then, for human cells, we also examined another protocol published by Rudolf Jaenisch's group to convert hESCs (H1-*EED*^{-/-}/*EED*-OE or H1-*SUZ12*^{-/-}/*SUZ12*-OE) into the naïve state. Again, we found that the naïve state H1-*EED*^{-/-}/*EED*-OE or H1-*SUZ12*^{-/-}/*SUZ12*-OE hESCs converted based on Jaenisch's protocol keep undifferentiated status without PRC2 function upon withdrawal of DOX, while H1-*EED*^{-/-}/*EED*-OE or H1-*SUZ12*^{-/-}/*SUZ12*-OE maintained in primed state were differentiated upon DOX withdrawal. Together, based on the experiments from different PRC2 components, different cell lines and naïve types, our data strongly demonstrate the differential requirement for PRC2 in primed and naïve pluripotent cells. We hope these new data would be satisfied. These new data are provided in new Figure 6 and Supplementary Figure 7.

Reviewers' Comments:

Reviewer #1:

Remarks to the Author:

The authors have made a number of changes and answered many of the comments. The English is still wobbly and in some cases ambiguous, even when it has been revised.

I think the paper is acceptable for Nature Comms.

Reviewer #2:

Remarks to the Author:

In their revised manuscript, the authors provide considered and comprehensive replies to all queries raised by both reviewers. Additional data on other PRC2 components, and another type of naive human cell, do much to strengthen the conclusions of the study. More information on the teratoma results, and experiments with other BMP inhibitors in the differentiation work, also provide convincing additions to the manuscript. The text has been extensively revised to clarify a number of points.